# High-resolution structure of a fish aquaporin reveals a novel extracellular fold

Jiao Zeng[1], Florian Schmitz[2], Simon Isaksson[3], Jessica Glas[2], Olivia Arbab[2], Martin Andersson[3], Kristina Sundell[4], Leif A Eriksson[2], Kunchithapadam Swaminathan[1], Susanna Törnroth-Horsefield[5], Kristina Hedfalk[2]

Aquaporins are protein channels embedded in the lipid bilayer in cells from all organisms on earth that are crucial for water homeostasis. In fish, aquaporins are believed to be important for osmoregulation; however, the molecular mechanism behind this is poorly understood. Here, we present the first structural and functional characterization of a fish aquaporin; cpAQP1aa from the fresh water fish climbing perch (*Anabas testudineus*), a species that is of high osmoregulatory interest because of its ability to spend time in seawater and on land. These studies show that cpAQP1aa is a water-specific aquaporin with a unique fold on the extracellular side that results in a constriction region. Functional analysis combined with molecular dynamic simulations suggests that phosphorylation at two sites causes structural perturbations in this region that may have implications for channel gating from the extracellular side.

## Introduction

Osmoregulation is a major challenge for fish because they are in direct contact with water and continuously need to compensate for passive water loss or uptake, depending on the environment (Evans, 2008). Among the fish species, the osmoregulation of the climbing perch (*Anabas testudineus*) is of particularly high interest because of being extremely adaptable to environmental changes. *A. testudineus* is a fresh water teleost that (i) can acclimate from fresh water to seawater (SW) over 7 d in a progressive manner (Chang et al, 2007), (ii) is able to live on land for up to 6 d and use air by breathing through the "labyrinth organ," a special accessory respiratory organ (ABO) located in the upper part of the gill chambers (Davenport & Abdulmatin, 1990), and (iii) is capable of active ammonia excretion during emersion (Tay et al, 2006; Ip et al, 2013) (Fig 1A). For these reasons, *A. testudineus* provides an attractive

model organism for studying the mechanisms of water homeostasis and ammonia excretion in fish.

Although the ionic osmoregulatory mechanisms and the structural organization of both gill and the intestinal epithelia are very well understood (Grosell, 2010; Sundell & Sundh, 2012; Lema et al, 2018), the mechanism for controlling water flow through aquaporins (AQPs) in various fish tissues and organs are less well studied (Sundell & Sundh, 2012; Madsen et al, 2015). AQPs are membrane-integral water channels that are present in all organisms where they facilitate passive flow of water across the membrane as a direct response to the osmotic pressure. AQPs in fish were first identified in 2000 and are distributed in the skin, gill, intestine, and kidney (Cutler, 2000). Interestingly, there seems to be a connection between the development of the fish and the diversity of AQPs produced, reflecting the trends of evolution (Tingaud-Sequeira et al, 2008), as illustrated by 18 and 42 AQP members having been identified in the model organism zebrafish *Danio rerio* (Tingaud-Sequeira et al, 2008) and the Atlantic salmon (*Salmon salar*) (Madsen et al, 2015), respectively. Furthermore, expression levels vary with changes in salinity supporting the involvement of AQPs in fish osmoregulation (Guo et al, 2017). Most of our current understanding of the osmoregulatory role of AQPs in fish concerns the fish intestine where they are found in both the apical and basolateral membranes of intestinal epithelial cells, mediating transcellular water flow and intestinal water uptake in SW (Fig 1B) (Fischbarg, 2010; Madsen et al, 2015). Investigation of numerous fish species reveals that AQPs are up-regulated in the intestine as a response to transfer to SW, indirectly supporting their involvement in a general mechanism for osmoregulation and water absorption (Sundell & Sundh, 2012; Madsen et al, 2015). The presence of AQPs in gills and skin suggest that they are also involved in regulated transmembrane water flow in these organs, but little is known about the osmoregulatory role of AQPs found in tissues and organs that are in direct contact with the surrounding (Madsen et al, 2015).

AQPs are commonly divided into three subgroups: orthodox AQPs, permeable to water only; aquaglyceroporins that also permit

[1]Department of Biological Sciences, National University of Singapore, Queenstown, Singapore  [2]Department and Chemistry and Molecular Biology, Gothenburg University, Göteborg, Sweden  [3]Department of Chemistry and Chemical Engineering, Applied Surface Chemistry, Chalmers University of Technology, Gothenburg, Sweden  [4]Department of Biology and Environmental Sciences, Gothenburg University, Göteborg, Sweden  [5]Department of Biochemistry and Structural Biology, Centre for Molecular Protein Science, Lund University, Lund, Sweden

Correspondence: susanna.horsefield@biochemistry.lu.se; kristina.hedfalk@gu.se

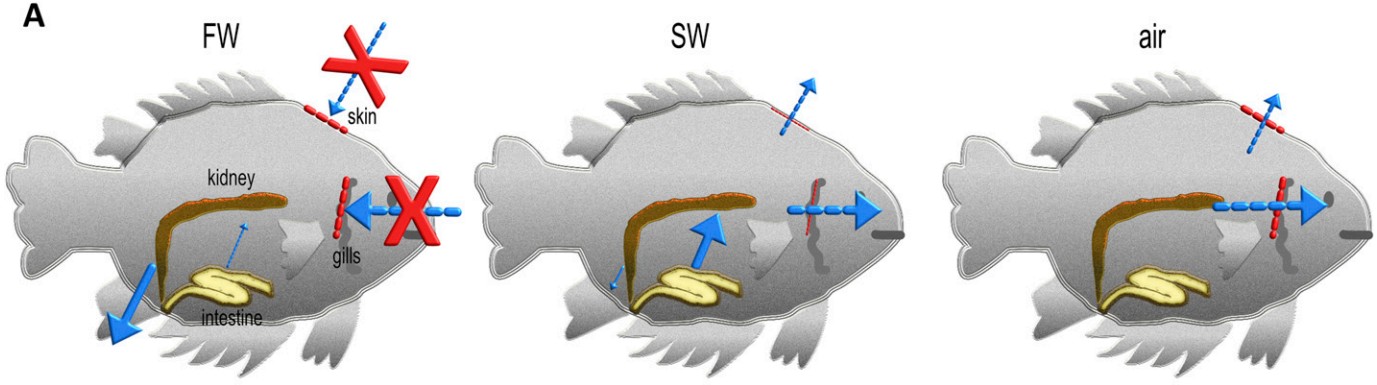

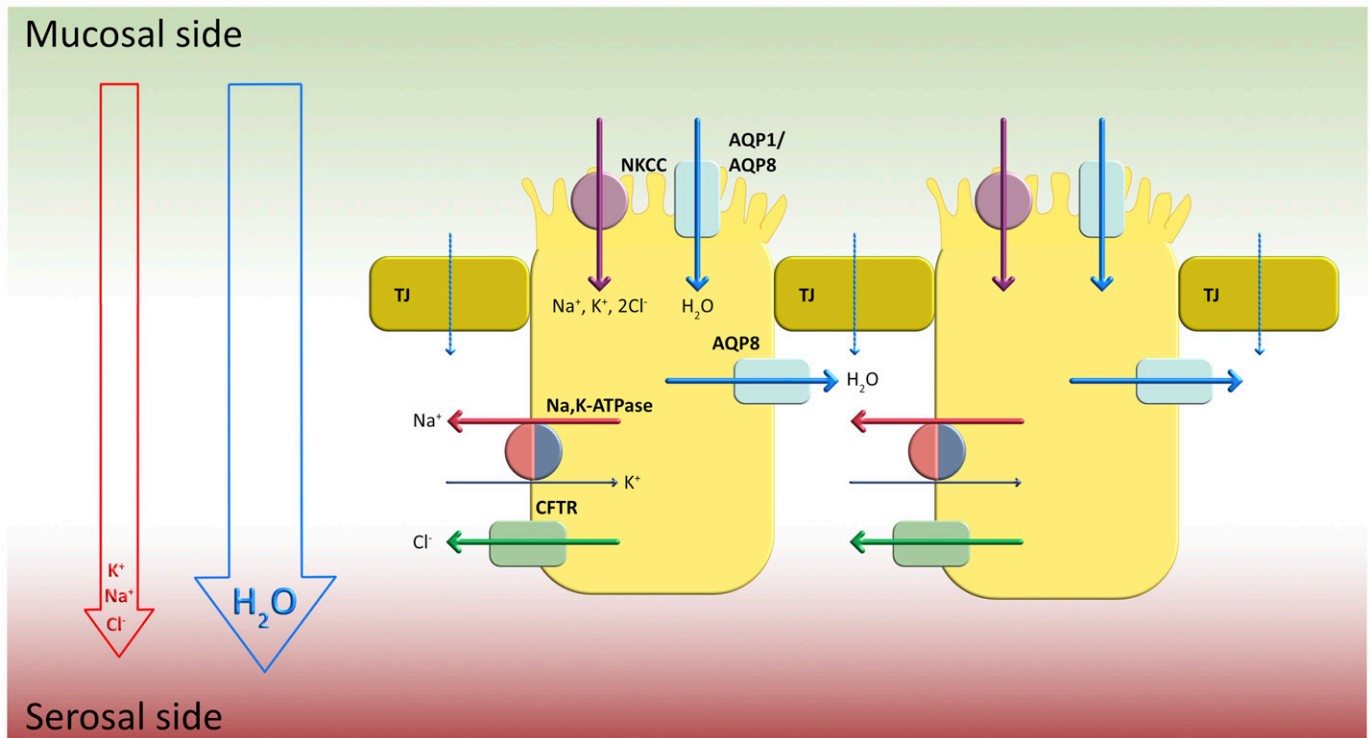

**Figure 1. Water passage in climbing perch and water flow in the fish intestine.**
**(A)** Schematic illustration of the assumed main water flow for climbing perch in fresh water, seawater (SW), and air, respectively, where the size of the arrow corresponds to the water flow and the tissue barrier is indicated by a dashed red line. In fresh water, *Anabas testudineus* is a hyperosmotic osmoregulator, aiming to minimize the diffusional inflow of water across the gills and producing dilute urine to get rid of excess water. In SW, on the other hand, *A. testudineus* will acclimate and become a hypoosmotic osmoregulator with the reverse problem with water diffusing out across all epithelia. This is counteracted by an ion-coupled water uptake across the intestine and a decreased glomerular filtration rate and urine production. During terrestrial exposure, *A. testudineus* must reduce water loss from the large surface areas constituted especially by the gills but also by the skin. The intestinal water absorption is also likely to be reduced as that is dependent on drinking, which ceases in air. **(B)** During SW acclimation, a total water flow is directed from the mucosal to the serosal side of the intestine driven by an ionic gradient of mainly $K^+$, $Na^+$, and $Cl^-$, involving ion transport via NKCC, Na,K-ATPase, and CFTR. In the SW trout, the increased intestinal water absorption is facilitated by aquaporin isoforms in the apical (AQP1a and AQP8ab) and basolateral (AQP8ab) membranes, respectively (reviewed in Madsen, 2015). The figure shows two enterocytes separated by tight junctions (TJ).

flow of larger solutes such as glycerol and urea; and super-aquaporins with unknown solute specificity (Ishibashi et al, 2009). In addition, some AQPs have been shown to allow passage of gas, most notably ammonia and carbon dioxide, and a selection of the

AQP isoforms allow flow of hydrogen peroxide as part of the cellular redox signaling (Nordzieke & Medrano-Fernandez, 2018). High-resolution structures of various AQP homologs have revealed a common homotetrameric fold, with each monomer constituting a

functional pore and comprising six transmembrane helices. In addition, two of the connecting loops form half membrane–spanning helices, each containing one copy of the highly conserved NPA (asparagine-proline-alanine) motif (Ozu et al, 2018) (Fig 2). Within the pore, the diameter and polarity of the aromatic/arginine (ar/R) motif located near the extracellular vestibule determine the substrate specificity of the AQP channel, with the pore diameter being restricted to ~3 Å in water-specific AQPs and typically being 1 Å wider in aquaglyceroporins (Fu et al, 2000; Sui et al, 2001). High-resolution structural studies have also explained how the NPA- and ar/R motifs together prevent protons being conducted along the single file of water molecules inside the pore via the Grotthuss mechanism (Tornroth-Horsefield et al, 2010; Eriksson et al, 2013). Eukaryotes have evolved to posttranslationally regulate water flow through AQPs by gating or trafficking in response to environmental or cellular signals. Gating has been proposed for several eukaryotic aquaporin homologs and includes restricting the channel radius by larger conformational changes (capping) or small side-chain movements (pinching) (Hedfalk et al, 2006). Structural studies have resulted in detailed gating mechanisms to be proposed for mammalian AQP0 (Gonen et al, 2004; Reichow et al, 2013), SoPIP2;1 from plant (Tornroth-Horsefield et al, 2006; Nyblom et al, 2009; Frick et al, 2013a, 2013b) and Aqy1 (Fischer et al, 2009) from yeast, involving triggers such as pH (AQP0 and SoPIP2;1), phosphorylation (SoPIP2;1 and Aqy1), $Ca^{2+}$-binding (SoPIP2;1), protein–protein interactions (AQP0), and mechanosensitivity (Aqy1).

To shed light on the osmoregulatory role of AQPs in fish, we have structurally and functionally characterized AQP1aa from *A. testudineus* (cpAQP1aa), a homolog of mammalian AQP1 and the only characterized AQP from this species to date (Ip et al, 2013) (Fig S1). Our work provides the first structural and functional characterization of a fish AQP and shows that cpAQP1aa is a water-specific AQP with a unique pore narrowing at the extracellular side. Mutational analysis combined with molecular dynamic (MD) simulations suggest that perturbations of this region may lead to conformational changes that could control the water flow through cpAQP1aa, the implications of which in relation to a possible extracellular gating mechanism is discussed. This study contributes significantly to our understanding of the role of AQPs in fish and suggests new intriguing possibilities for the variation in molecular mechanisms that control water flow through the AQP family of proteins.

## Results

### High-resolution structure of cpAQP1aa reveals a novel fold on the extracellular side

Because flexible termini are known to hamper crystallization of AQPs, C-terminally truncated cpAQP1aa (cpAQP1aa-243, where residues 243–261 were deleted) was produced in *Pichia pastoris*, purified, and crystallized. The structure was solved to 1.9 Å resolution by molecular replacement using a homology model of cpAQP1aa based on the crystal structure of bovine AQP1 (PDB code 1J4N) (Sui et al, 2001) (Table S1). The overall structure of cpAQP1aa shares most of the common AQP structural features; assembling as a homotetramer

with each monomer forming a functional pore and consisting of six transmembrane α-helices (1–6) and five connecting loops (A-E) (Fig 2A and B). A seventh pseudo-transmembrane segment is formed by the insertion of two half-helices in loops B and E into the pore, each of which contain the AQP signature NPA motif. A single file of water molecules is aligned within the pore of each monomer, indicating that cpAQP1aa is in a functional state. Loops A and C are partly disordered, with residues 34–37 and 108–112, lacking electron density, and these parts have been omitted from the structure. The last residue that could be unambiguously modeled is Lys 226, wherefore its C-terminus is deemed to be disordered beyond this residue.

To explore the features of the cpAQP1aa pore, its shape and dimensions were determined using the program HOLE (Smart et al, 1993) (Fig 2C). Similar to other AQPs, the cpAQP1aa pore is dumbbell-shaped, consisting of open vestibules at both intracellular and extracellular sides, with an elongated narrow pore in between. The pore constricts to a radius of ~1.2 Å at the ar/R region, formed by residues Arg 187, His 172, Cys 181, and Phe 50, a width and residue composition that is typical for water-specific AQPs (Sui et al, 2001). However, when comparing the channel profile of cpAQP1aa with those of the water-specific channel human AQP4 (Ho et al, 2009), the aquaglyceroporin human AQP7 (de Mare et al, 2020), and the ammonia-permeable TIP2;1 from *Arabidopsis thaliana* (Kirscht et al, 2016), a unique pore narrowing at the extracellular side is identified. This region is defined by a novel conformation of loop C, placing residues 114–117 (cpAQP1aa numbering) closer to the center of the extracellular vestibule. As a result, Leu 117, Ile 176, and Gly 180 form a hydrophobic constriction region with a radius of 1.1 Å, ~8 Å away from the ar/R region. The unique conformation of loop C is stabilized by a hydrogen-bonding network that involves Tyr 107 in transmembrane helix three, residues 114–119 in loop C, and Arg 187 within the ar/R region (Fig 2D). Specifically, hydrogen bonds are formed between the side chain of Tyr 107 and the backbone nitrogen of Leu 119 as well as the carbonyl oxygen and side chain of Asn 114 and 116, respectively, via a water molecule. In addition, the Arg 187 guanidino group forms hydrogen bonds to the backbone oxygens of Leu 117 and Gly 118.

A structural comparison between cpAQP1aa, human AQP4 (water-specific), and human AQP7 (glycerol-permeable) (Fig 2E) as well as an extended comparison with several other eukaryotic AQPs (Fig S2) shows structural variability in loops A and C, highlighting that these two loops are structurally not well conserved. Although Tyr 107 is conserved in all three AQPs, its side chain is pointing away from the channel in AQP4 and AQP7, resulting in loop C remaining further away from the channel opening (Fig 2F). Furthermore, Leu 117 is highly conserved in vertebrate AQPs, including fish and human (albeit not in human AQP7) (Fig S1), indicating that the extracellular constriction region is not directly related to this residue being present or not. It should be pointed out that despite the restricted pore diameter, this region is not narrow enough to completely exclude water molecules and additional conformational changes would be needed to block water flow.

### The extracellular fold of cpAQP1aa is unaffected by pH

To investigate if the conformation of loop C that is responsible for the cpAQP1aa extracellular constriction region is dependent on pH,

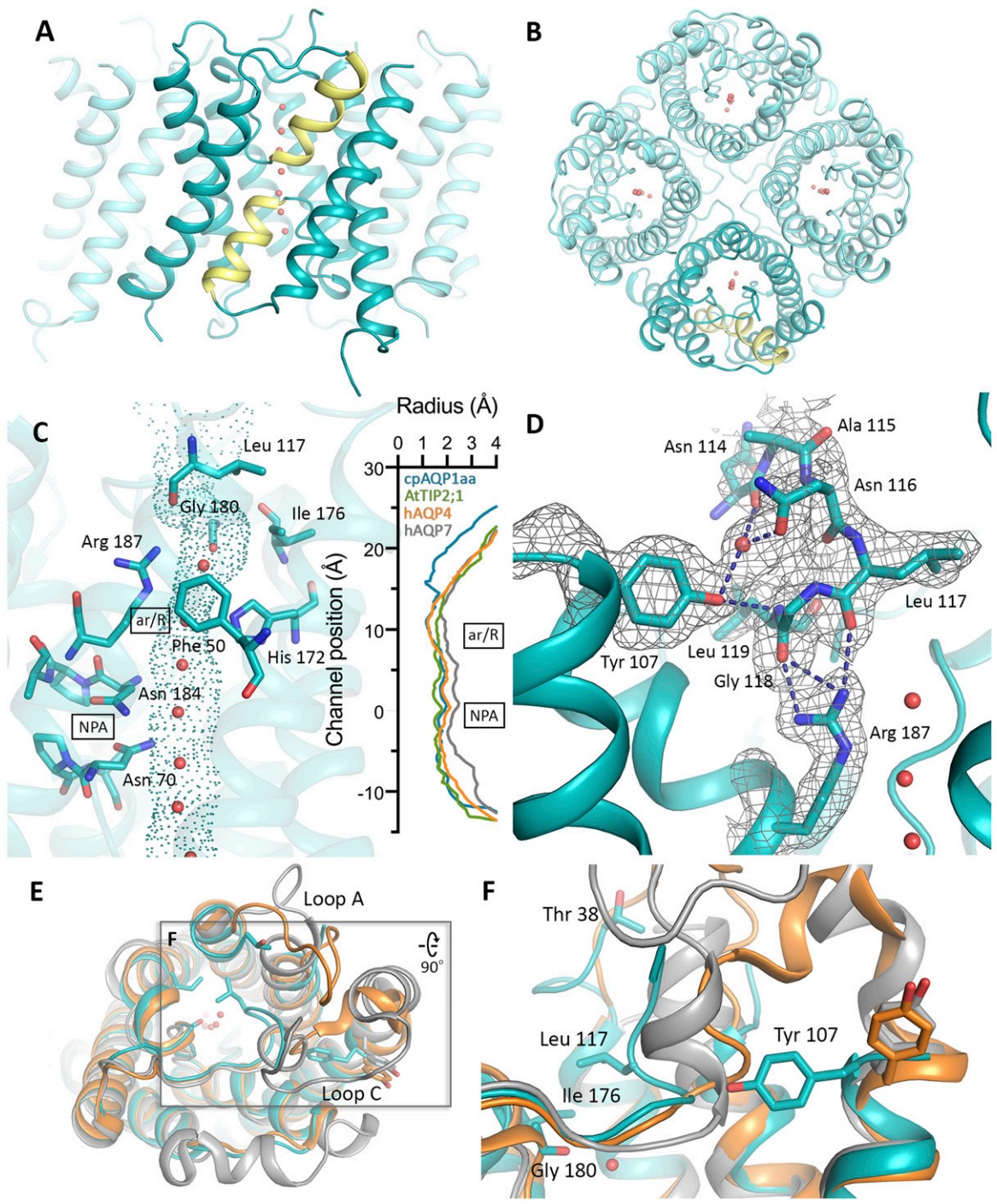

**Figure 2.  Crystal structure of cpAQP1aa.**
**(A, B)** The cpAQP1aa tetramer viewed (A) parallel to the membrane and (B) from the extracellular side. The two half-helices forming the seventh pseudo-transmembrane segment are colored yellow in the frontmost monomer. Water molecules inside the channel are showed as red spheres. **(C)** HOLE analysis of the water-conducting channel with the channel dimensions visualized as dots. Residues in the aromatic-arginine (ar/R) and NPA regions as well as the extracellular vestibule are shown in stick representation. The graph shows a comparison between the channel profiles of cpAQP1aa (teal), human AQP4 (orange, PDB code 3GD8), human AQP7 (gray, PDB code 6QZI), and *Arabidopsis* TIP2;1 (green, PDB code 5I32). The zero channel position corresponds to the midpoint between the two NPA motifs. For cpAQP1aa, the channel is significantly narrower on the extracellular side of the ar/R region with a channel radius of 1.1 Å around Leu 117 (~16 Å from the NPA region). **(D)** Close-up view of the hydrogen-bonding network involving Tyr 107, Arg 187, and residues 114–119 that stabilizes the unique conformation of loop C in cpAQP1aa. Hydrogen bonds are depicted as blue dashed lines. Electron density shown as gray mesh represents the 2FoFc-map contoured at 1.0 $\sigma$. **(E)** Structural overlay of cpAQP1aa (teal), human AQP4 (orange), and human AQP7 (gray), viewed from the extracellular side, showing that the main structural differences are found in loops A and C. Residues in the cpAQP1aa extracellular constriction region and putative phosphorylation site at Thr 38 in loop A are shown in stick representation. Water molecules in the channel are shown as red spheres. **(E, F)** Zoom-in on boxed region in (E), viewed from the side of the membrane (90 degree rotation), showing significant structural differences in loop C between

we crystallized and solved the structure of cpAQP1aa at a lower pH; pH 6.5 instead of 7.8. The low pH crystals belonged to a different space group, C222$_1$ instead of P42$_1$2, and diffracted to significantly lower resolution (3.5 Å) (Table S1). The lower diffraction quality is consistent with the differences in crystal packing; the crystals grown at the lower pH have considerably weaker crystal contacts and higher solvent content compared with the crystals at pH 7.8 (Fig S3). Furthermore, in the structure at lower pH, there are four cpAQP1aa monomers in the asymmetric unit compared with one in the structure at higher pH. Because of this, structural differences can be observed between the monomers, most notably the fact that loops A and C could be built in their entirety in one of the monomers (monomer C) (Fig S4A and B).

As seen in Fig S4C and D, the two structures at different pH are very similar, although some minor differences can be seen in loops A and C. Nevertheless, the part of loop C that allows the formation of the extracellular constriction region (residues 114–119) occupies a similar position at both high and low pH. In particular, Tyr 107, Asn 116, Leu 117, Gly 118, and Arg 187 line up well and seem to be able to participate in the same hydrogen-bonding network in both structures (hydrogen bonds cannot be reliably assigned in the low pH structure because of the limited resolution) (Figs 2D and S4D). The channel profiles, as calculated by HOLE, are very similar in both structures (Fig S4E), suggesting that the extracellular hydrophobic constriction region is unaffected by pH.

### cpAQP1aa is a channel for water but not glycerol

To verify that the cpAQP1aa produced in *P. pastoris* is a functional water channel, as suggested by the crystal structure, both the full-length protein (cpAQP1aa-FL) and the truncated variant used for crystallization (cpAQP1aa-243) were reconstituted in liposomes and tested for water flow by stopped-flow spectroscopy (Fig S5A). The rate of water flow was compared with that of human AQP4 (hAQP4) and the empty liposomes. The initial rate constants were 37.59 ± 13.84 s$^{-1}$ and 39.97 ± 11.92 s$^{-1}$ for cpAQP1aa-FL and cpAQP1aa-243, respectively (Fig S5B). This is significantly higher than that for the empty liposome (9.49 ± 3.68 s$^{-1}$), showing that both cpAQP1aa constructs are equally permeable to water. However, compared with the initial rate constant for hAQP4 (95.13 ± 31.67 s$^{-1}$), a highly efficient water-specific AQP with a fully open pore, the rate constant of cpAQP1aa is substantially lower. This is in agreement with the cpAQP1aa pore being restricted but not closed, as observed in the crystal structure (Fig 2A and B). As seen in Fig S6A, the liposomes contain similar levels of protein, showing that the differences in water flow rates between hAQP4 and cpAQP1aa are not a result of liposome reconstitution efficiency.

Because many AQPs have broader specificity than water, glycerol permeability was also evaluated. Neither of the reconstituted AQPs, cpAQP1aa-FL, cpAQP1aa-243, or hAQP4, permitted flow of glycerol (Fig S5C). Similarly as for the water flow assay, immunoblot analysis supports that the AQPs are properly reconstituted, with cpAQP1aa being reconstituted at slightly higher levels than hAQP4 (Fig S6B).

The lack of glycerol flow through cpAQP1aa is well related to the observed diameter and structure of the pore, which is significantly narrower than the glycerol facilitator human AQP7 (Fig 2C).

### The structural features of the cpAQP1aa pore do not support passage of ammonia

To explore if the cpAQP1aa pore could permit passage of ammonia, we compared its structural features to those of the ammonia channel PIP2;1 from *A. thaliana* (AtPIP2;1). The ammonia channel AtPIP2;1 has been described as comprising an extended specificity filter, which, together with a deprotonation mechanism provided by His 131, is believed to contribute to ammonia passage, in addition to water (Kirscht et al, 2016). The presence of a histidine in position 131 forces the side chain of the arginine in the ar/R region of AtTIP2;1 to adopt a novel conformation that has not been seen in other AQPs (Fig 3). Furthermore, the ammonia permeability of AtTIP2;1 has been proposed to correlate with the conformation of Gly 194, for which the carbonyl position and hydrogen-bonding pattern is typical for non–water-specific AQPs. When comparing this region in AtTIP2;1 and cpAQP1aa, it is evident that the structure of cpAQP1aa most strongly resembles those of water-specific AQPs. Specifically, the presence of an asparagine (Asn 120 in cpAQP1aa) in the equivalent position to His 131 in AtPIP2;1, the position of the cpAQP1aa Cys 181 carbonyl (equivalent position as Gly 194 in AtTIP2;1), and the hydrogen-bonding pattern between residues and water molecules all point at cpAQP1aa being a water-specific AQP. This suggests that it is rather unlikely that cpAQP1aa plays a major role in ammonia transmembrane passage as previously proposed based on mRNA analysis (Ip et al, 2013).

### Mutational analysis of the extracellular constriction region

The structure of cpAQP1aa reveals that loop C forms a unique conformation that restricts the pore at the extracellular side and is stabilized by a hydrogen-bonding network located near Leu 117, specifically involving residues Tyr 107, residues 114–119, and Arg 187 (Fig 2C and D). To evaluate the importance of Leu 117 and Tyr 107 for cpAQP1aa water flow, we mutated these residues to alanine and studied their water permeability in liposomes. As seen in Fig 4A and Table S2, proteoliposomes containing L117A had a similar water flow rate as wild-type cpAQP1aa (normalized values), suggesting that it is not the side chain of Leu 117 in itself that is critical for the restricted water flow of cpAQP1aa but rather the overall fold of the loop C backbone. This is in agreement with the structural analysis, which showed that the narrowest point of the channel is defined by the backbone carbonyls of Leu 117 and Gly 180 and not the Leu 117 side chain (Fig 2C). In contrast, Y107A showed a significantly higher water flow rate (normalized values), in the range of that of hAQP4, suggesting this mutation gives rise to a fully open pore (Fig 4A and Table S2). This strongly supports that the tyrosine side chain is important for forming the network that stabilizes the unique fold of loop C seen in cpAQP1aa.

the three overlaid structures. In cpAQP1aa, the side chain of Tyr 107 has flipped compared with AQP4 and AQP7, pushing loop C further into the extracellular vestibule closer to the channel opening.

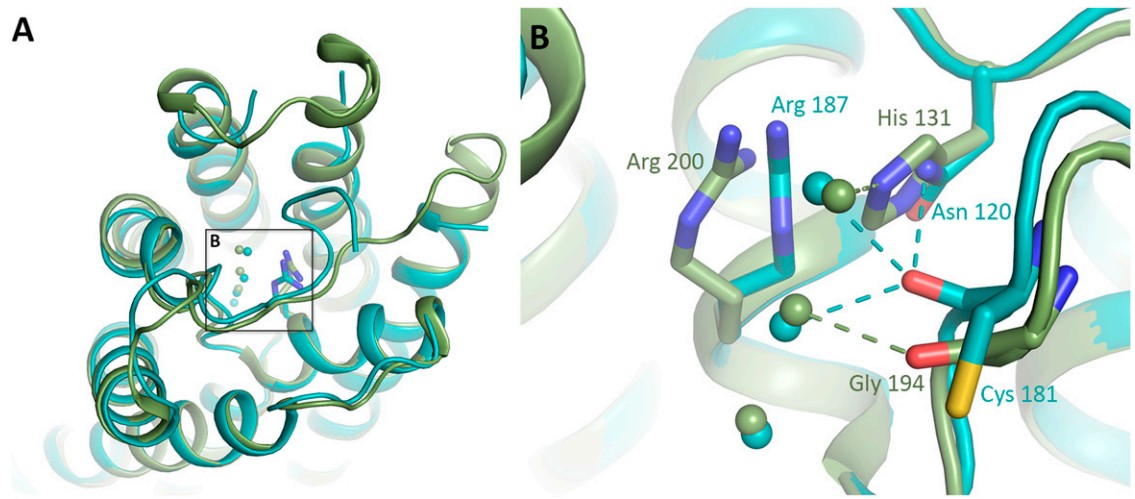

**Figure 3. Structural comparison between cpAQP1aa and the ammonia-facilitating aquaporin AtTIP2;1.**
**(A)** Overlay of cpAQP1aa (teal) and AtTIP2;1 (green) viewed from the extracellular side. The arginine in the ar/R region is shown in stick representation. Water molecules in the channel are shown as spheres of the respective protein color. **(A, B)** Zoom-in on the boxed area in (A), showing the different side-chain orientation of the arginine in the ar/R region and the hydrogen-bonding pattern between water molecules and residues within the channel. Hydrogen bonds are shown as dotted lines in the respective protein color. In cpAQP1aa, the carbonyl position of Cys 181 and its hydrogen bonds to Asn 120 and two water molecules correspond to what is typically seen in water-specific AQPs. In contrast, the carbonyl of Gly 194 in AtTIP2;1 occupies a different position, observed mainly in non–water-specific AQPs, and shows a different hydrogen-bonding pattern.

To identify mutations that may be of further relevance for controlling cpAQP1aa water flow, we analyzed the cpAQP1aa sequence putative phosphorylation sites using the GPS5.0 server (Wang et al, 2020). Two sites of potential interest were identified: Tyr 107 and Thr 38, the latter which is located in loop A, in close vicinity to the loop C constriction region (Fig 2F). We hypothesized that phosphorylation of these two sites could trigger conformational changes of loop C that affect the pore dimensions. To explore this, we created alanine and glutamate mutants of both sites, a common strategy to abolish and mimic phosphorylation, respectively, and assayed their water permeability (Fig 4A, Table S2, and Fig S6C and D). As described above, the Y107A displayed increased water permeability. This was most likely because of the loss of the hydrogen bond between Tyr 107 and loop C. For Y107E, however, the water permeability was similar to wild-type cpAQP1aa. This is consistent with glutamate generally being considered as a poor mimetic of a phosphorylated tyrosine, as opposed to phosphorylated threonine and serine (Anthis et al, 2009). In contrast, both T38A and T38E were significantly more efficient than wild-type (but not significantly different from each other), facilitating water passage at a similar rate as hAQP4 and Y107A. This suggests that mutation of this specific residue could lead to conformational changes that result in altered pore dimensions.

To rule out that our results were not influenced by cpAQP1aa being phosphorylated by the expression host, as shown previously for AQPs expressed in *P. pastoris* (Ampah-Korsah et al, 2016; Roche et al, 2017), we investigated the phosphorylation status of recombinant cpAQP1aa. First, the protein construct used for crystallization (cpAQP1aa-243) was analyzed by immunoblot using a phosphate-binding tag to identify phosphorylated proteins. The blot showed that cpAQP1aa-243 is indeed phosphorylated (Fig S7). Although the electron density map clearly shows that Tyr 107 is not

phosphorylated in the crystal structure (Fig 2D), the phosphorylation status of Thr 38 is more difficult to evaluate because the electron density for this residue is much less clear. Moreover, it cannot be fully ruled out that a fraction of the protein in solution is phosphorylated on either residue but not included in the crystal. We therefore further analyzed the phosphorylation status of full-length cpAQP1aa using mass spectrometry. Phosphorylation was confirmed at three C-terminal residues, Ser 239, Thr 254, and Thr 255 (Fig S1), but despite multiple efforts, no phosphorylation was found on either Thr 38 or Tyr 107. This shows that neither of the constructs used for structural and functional studies are phosphorylated at the two sites proposed to influence the conformation of loop C (Thr 38 or Tyr 107) but may be phosphorylated at sites within the C-terminus that are not visible in the structure because of disorder (Ser 239) or truncation (Thr 234 and 255). As such, we are confident that the functional characterization of wild-type cpAQP1aa rather than their respective alanine mutation gives the most accurate estimate the water transport rate through cpAQP1aa that is not phosphorylated on Tyr 107 or Thr 38.

**MD simulations of loop C**

To shed additional light on the influence of mutations and putative phosphorylation on the conformation of loop C in cpAQP1aa, MD simulations were applied to explore the roles of Thr 38, Tyr 107, and Leu 117. For this purpose, the phosphorylated variants pT38 and pY107 as well as the L117A, Y107A, and Y107E mutants were generated. Noteworthy, all simulated structures are highly stable in the time frame of the current simulations (100 ns) (Fig S8). Similar to the crystal structure (Fig 2D), the MD simulations show a clear role for Tyr 107 and Arg 187 in stabilizing the conformation of loop C that is responsible for forming the extracellular constriction region in

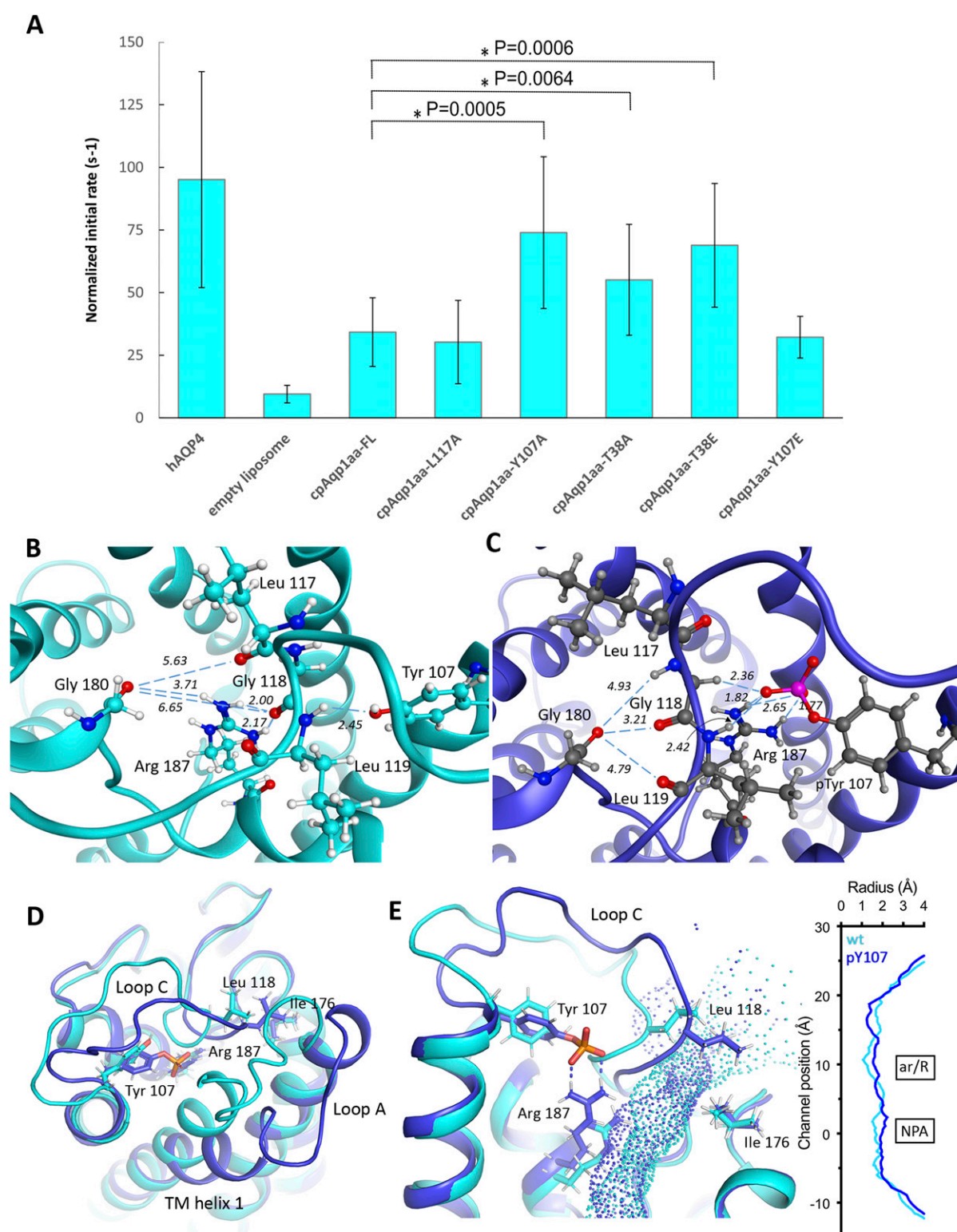

**Figure 4. Mutational and in silico analysis of the extracellular constriction region.**
**(A)** The initial water permeability rates for wild-type and mutant cpAQP1aa with the rate of hAQP4 shown for comparison. All values are normalized taking the protein amount into account (n = 6–31 ± SD). The normalized initial water flow rates show that the Y107A, T38A, and T38E mutants are more efficient than the wild-type protein, in the range of hAQP4 (no significant differences), whereas the L117A and Y107E are similar to wild-type. **(B)** Zoom-in the pore opening in an energy-minimized final snapshot from the 100-ns molecular dynamic simulation of the wild-type system with Tyr 107, Leu 117, Gly 118, Leu 119, Gly 180, and Arg 187 highlighted. Distances are given in Ångström. **(B, C)** Same as (B) but for the pY107 system. The extracellular loop C has shifted to the left, narrowing the pore opening, and is held in place by the interactions

cpAQP1aa. Specifically, in the simulated wild-type cpAQP1aa structure, hydrogen bonding is established between the phenolic oxygen of Tyr 107 and the backbone NH of Gly 118 (2.45 Å in the minimized structure after 100-ns MD simulation), and two hydrogen bonds are formed between Arg 187 and the carbonyl oxygen of Gly 118 (distances 2.00 and 2.17 Å) (Fig 4B). In addition, a stable interaction between Arg 198 and the carbonyl of Gly 180 is formed (O - - HN distance 3.71 Å). The distances between the backbone carbonyl oxygen of Gly 180 and those of Leu 117, Gly 118, and Leu 119, representing the pore opening, remain very stable throughout the simulation (5.63, 6.65, and 6.82 Å in the minimized final snapshot, Fig 4B). These values are highly similar values to those obtained for pT38, L117A, Y107A, and Y107E, suggesting that they are in a similar structural state. In contrast, for the pY107 variant, the system gradually adjusts to the bulkiness, altered electrostatics, and hydrogen-bonding conditions of the phosphorylated side chain. During the initial stages of the simulation, Gly 118 is pushed toward Gly 180 by the bulky phosphate group of pY107 (Fig 4C). In addition, the backbone of Leu 117 rotates so that its carbonyl points away from the pore opening and the amine group of Gly 118 instead orients toward Gly 180. After about 50 ns of the production run, there is a rotation of the phosphate group which in turn is accompanied by a motion of Arg 187 from its position near the pore opening, to instead forming two strong hydrogen bonds to the phosphate of pY107. The phosphate group also forms strong interactions to the backbone NH of Leu 119 and a hydrogen of Gly 118. Because of the shift in the loop C orientation, kept in place by the strong interactions to the pY107 – R187 pair, the distance between the carbonyl oxygen of Gly 180 and Gly 118 is shortened to 3.21 Å (and 4.8–4.9 Å to the CO of Leu 119 and NH of Leu 117, respectively). These distances are significantly shorter than the 5.6–6.9 Å observed in the wild-type system. Moreover, to accommodate for the change in the position of Arg 187, the distal part of transmembrane helix 1 is displaced ~ 6 Å compared with the simulated wild-type system, causing loop A to move away (Fig 4D).

To explore the effect of phosphorylation on the pore dimensions, we analyzed the coordinates from the final stage of the wild-type and pY107 simulations using HOLE (Fig 4E). For wild-type cpAQP1aa, the constriction region is somewhat wider than in the crystal structures. This suggests that there is some room for flexibility in this region and that wild-type cpAQP1aa may be somewhat less restricted in solution than in the crystal structure. In contrast, the shift in the position of loop C in pY107 causes the pore to narrow to a radius of ~1.2 Å, a similar value as that seen in the crystal structure (Fig 2C). In addition, the pore widens at the ar/R region because of the arginine being pulled away from the pore center by its interaction with the Tyr 107 phosphate group, adopting a pore diameter more similar to aquaglyceroporins. These structural changes around the constriction and ar/R regions suggest that phosphorylation of Tyr 107 could lead to altered pore dimensions and/or solute permeability.

## Discussion

The current understanding of the physiological relevance of AQPs in fish is mainly based on gene identification and phylogenetic analysis and has suggested that fish AQPs are involved in many physiological functions including osmoregulation and passage of gas such as ammonia and $CO_2$ (Horng et al, 2015; Talbot et al, 2015). However, the molecular details of fish AQP function and regulation, and hence their role in fish physiology, have been lacking. In the current study, we present the first structural and biochemical analysis of a fish AQP, cpAQP1aa, from the climbing perch, an AQP that has been previously proposed to have its main role in ammonia excretion (Ip et al, 2013). Our studies show that cpAQP1aa is a water-specific AQP that is not permeable to glycerol (Fig S5), comprises a residue composition at the ar/R region that is consistent with water specificity (Fig 2), and lacks the structural features that are believed to be associated with ammonia passage (Fig 3). Given its high expression in gills and skin, tissues that are in direct contact with the fish surroundings (Ip et al, 2013), it therefore seems plausible that cpAQP1aa is involved in osmoregulation.

The most striking feature of the cpAQP1aa structure is the presence of a constriction region at the extracellular side of the pore formed by a unique conformation of loop C (Fig 2C and D). From a sequence comparison, it is not obvious why the conformation of loop C differs in cpAQP1aa compared with other structurally characterized AQPs (Fig S1). A possible explanation lies in the hydrogen-bonding network surrounding Leu 117 which, through interactions with Tyr 107, stabilizes loop C in its unique conformation (Fig 2D). Interestingly, Tyr 107 is rather well-conserved among fish species and present in several mammalian AQP homologs, including AQP0, AQP4, and AQP7 (Fig S1), However, in mammalian AQPs, the tyrosine side chain adopts a different position, pointing away from the pore rather than toward it (Fig 2F). Furthermore, in cpAQP1aa, Tyr 107 is followed by a glycine, a residue that is highly conserved in fish but varies in other eukaryotic AQPs (Fig S1). It may be that the combination of tyrosine and glycine at these positions is necessary for the unique conformation of loop C seen in cpAQP1aa. Furthermore, the conservation of the Tyr–Gly motif in fish suggests that the extracellular constriction region observed in cpAQP1aa may be a general feature among fish AQPs.

In addition to restricting the diameter of the pore, the conformation of loop C also allows for the formation of hydrogen bonds between Arg 187 of the ar/R region and the loop C backbone, specifically the carbonyls of Leu 117 and Gly 118. A similar interaction between the arginine and loop C was also seen in the crystal structure of the AQP from the malaria parasite *Plasmodium falciparum* (PfAQP) (Fig S9) (Newby et al, 2008). Through this interaction, the hydrogen-bonding sites of the arginine guanidino group are saturated, a feature that is typically seen in water-specific AQPs but not aquaglyceroporins. Based on mutational and structural analysis of PfAQP, it was proposed that the hydrogen-bonding

between Arg 187 and phosphorylated Tyr 107. **(D, E)** Overlay of the final snapshots from the simulations for wild-type (cyan) and pY107 (blue). In pY107, Arg 187 is pulled away from the pore center because of its interaction with the phosphorylated tyrosine side chain, with its new position causing the distal end of transmembrane helix 1 and loop A to be displaced. **(E)** Analysis of the pore dimensions using HOLE (E) shows that the constriction region (channel position ~15 Å) is narrower in pY107 than in the wild-type system. In addition, the new position of Arg 187 results in a widening of the ar/R region in pY107.

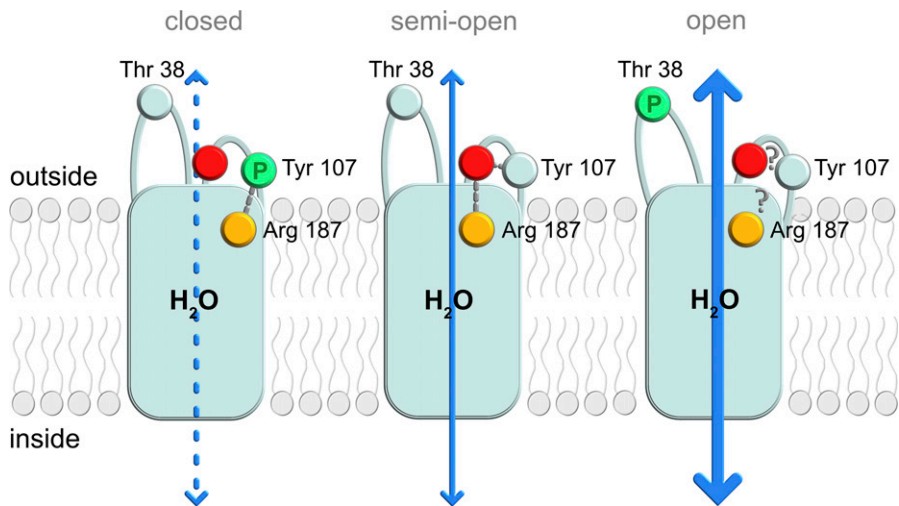

**Figure 5.  Proposed regulatory mechanism for cpAQP1aa.**
Schematic of a possible extracellular gating mechanism where phosphorylation of Thr 38 and Tyr 107 controls the water permeability. The crystal structure represents the non-phosphorylated state (middle) in which a hydrogen-bonding network between Tyr 107, backbone residues in the distal part of loop C (represented by a red sphere), and the ar/R region arginine (Arg 187, orange sphere) results in an extracellular constriction region that is consistent with a gate caught in a semi-open state. Upon phosphorylation of Tyr 107 (left), this hydrogen-bonding network is rearranged, and the phosphorylated tyrosine side chain instead interacts with Arg 187. This causes structural changes in this region, including further narrowing of the constriction region that is indicative of pore closure and a widened ar/R region. Phosphorylation of Thr 38 (right) results in an open pore; however, because of the structural flexibility of this region, a possible structural mechanism for this remains elusive.

pattern of the arginine in combination with a wide ar/R region underlies its unusual ability of being permeable to both glycerol and water at a high rate (Beitz et al, 2004; Newby et al, 2008). Given that the conformation of loop C allows similar interactions to occur also in cpAQP1aa, it may be that it also plays a role in solute selectivity and the permeability rate.

Although the conformation of the cpAQP1aa loop C does not cause complete pore closure, it is possible that a trigger such as phosphorylation or pH could cause structural changes of the loop that modulate water flow, as seen for SoPIP2;1 (Tornroth-Horsefield et al, 2006). In this respect, it is interesting to note the presence of two putative phosphorylation sites, Tyr 107 and Thr 38, within this region (Fig 2F). We hypothesize that the crystal structure of cpAQP1aa is caught in a semi-open state and that, similarly as for SoPIP2;1, the conformation of loop C could be subject to phosphorylation-induced changes that result in gating from the extracellular side (Fig 5). Although gating of cpAQP1aa remains to be shown in vivo, there are several key findings presented here that support this hypothesis. The water permeability of cpAQP1aa is similar to that of the gated PIP2;1 from spinach (SoPIP2;1), which has been shown to only reach its full capacity when locked in an open state by a phospho-mimicking mutation (Nyblom et al, 2009). This is also the case for cpAQP1aa, for which mutation of a putative phosphorylation site at Thr 38 in extracellular loop A increases the water permeability of cpAQP1aa to a similar level as that of hAQP4 (Fig 4A). Thr 38 is not a conserved residue among fish species (Fig S1) but appears in the frog, which opens up for speculations on a general osmoregulatory mechanism for an organism living both on land and in water. In the crystal structure, Thr 38 is located in a highly flexible region with less well-defined electron density, wherefore its exact position and conformation is uncertain. This may explain why MD simulations failed to capture any phosphorylation-dependent structural changes within the simulated timescale. Hence, a structural mechanism for how phosphorylation of Thr 38 may result in an open pore remains elusive.

In contrast to Thr 38, the second putative phosphorylation site in this region, Tyr 107, is highly conserved in fish (Fig S1), and its key role in stabilizing loop C in its unique position is clear (Figs 2D and

4A). Moreover, the MD simulations suggests significant structural perturbations when this residue is phosphorylated (Fig 4D). Although phosphorylation of Tyr 107 did not result in a fully closed pore, the observed structural changes results in pore narrowing as compared with the wild-type system (Fig 4E and F). It may be that this represents the initial structural events that follow Tyr 107 phosphorylation and that the transition to a fully open pore is not caught within the time frame of the simulations. The additional observation of a widened ar/R region consistent with an aqua-glyceroporin is intriguing and indicates a possible effect also on solute selectivity. However, because of the narrowness of the constriction region, further structural changes would be needed to allow glycerol to permeate the pore.

Extracellular gating has previously been proposed for mammalian AQP0 found in junctions in the eye lens fiber cells, for which a pinching-type mechanism was proposed. This was based on the observation that the Met 176 side chain protrudes into the channel in the junctional (closed) form (Gonen et al, 2005) (Fig S10A and B). As seen in Fig S1, Met 176 in AQP0 is a unique residue at this position among the AQPs; most isoforms have isoleucine at this site. In cpAQP1aa, this residue corresponds to Ile 176, which together with Leu 117 and Gly 180 forms the putative extracellular gate (Fig S10B and C). However, in comparison to mammalian AQP0, the proposed cpAQP1aa gating mechanism is more consistent with capping through a conformational change of loop C. As such, gating of cpAQP1aa would be more similar to that of SoPIP2;1 (Tornroth-Horsefield et al, 2006) for which a conformational change of a loop brings a leucine residue closer to the center of the pore end, albeit at the opposite side of the membrane (Figs 2 and S10C–E).

The possible involvement of extracellular phosphorylation sites in the proposed cpAQP1aa gating mechanism is intriguing. All other known phosphorylation-dependent AQP gating mechanisms involve cytoplasmic phosphorylation sites (Nesverova & Tornroth-Horsefield, 2019), and to our knowledge, this is the first time extracellular phosphorylation has been proposed to control gating of any membrane channel. Nevertheless, extracellular phosphorylation is well established, and several extracellular kinases have been characterized in mammals (Kinoshita et al, 2009; Bordoli et al, 2014; Cui

et al, 2015; Tagliabracci et al, 2015). For example, the secreted tyrosine kinase vertebrate lonesome kinase (VLK) has been shown to influence the localization and regulation of a synaptic receptor, supporting the role of extracellular phosphorylation in controlling membrane protein function (Hanamura et al, 2017). These extracellular kinases, including VLK, are also present in *A. testudineus*. Furthermore, VLK has been shown to be able to phosphorylate sequences that are highly similar to the sequence around Tyr 107 (Fig S11) (Bordoli et al, 2014), supporting this as a likely kinase candidate. In the case of Thr 38, it is more difficult to identify the most likely candidate out of several secreted Ser/Thr kinases, but a member of the recently identified secreted kinase family FAM69 (DIA1) is predicted to phosphorylate the same sequence (Ser Thr Pro) and is one possibility (Dudkiewicz et al, 2013; Hareza et al, 2018). The fact that neither VLK nor DIA1 is found in yeast could explain why cpAQP1aa was not phosphorylated at these sites by the expression host.

To conclude, the special characteristic of the climbing perch is its ability to both be truly euryhaline and be able to spend time on land and in air. Under these conditions, it is likely that the presence of gated AQPs is beneficial in order for the fish to finely regulate the water flow across its epithelia. Our structural and functional characterization of an aquaporin from this unique fish species has given novel insights into fish water homeostasis, including possible adaptation mechanisms that allow fish to withstand the osmolarity changes that arise from these large environmental variations.

# Materials and Methods

### Cloning and overproduction

The full-length gene for *A. testudineus*, climbing perch, aquaporin 1aa (cpAQP1aa-FL, residues 1–261) (Genbank accession code JX645188.1) was codon-optimized for protein production in *P. pastoris* (Genscript), including GCT after the first methionine to meet the yeast consensus sequence (Romanos et al, 1992) and subcloned into the pPICZA vector (Invitrogen), fused with a C-terminal 8xHis-tag (cpAQP1aa-FL). Two truncated constructs, untagged-cpAQP1aa-243 (residues 1–243) and C-terminal 8×His-tagged-cpAQP1aa-243, were prepared for crystallization. The genes were ligated into the pPICZA vector between the *EcoR*I and *Xho*I or *Not*I restriction sites, respectively. The plasmids were linearized by the restriction enzyme *Mss*I (Thermo Fisher Scientific) and transformed into an aquaporin-deficient Δ33 strain of *P. pastoris* by electroporation (Fischer et al, 2009). For all the constructs of cpAQP1aa used in this study, extra nucleotides "GCT" encoding an alanine were intentionally added after the start codon "ATG" to meet the Kozak consensus sequence (Öberg et al, 2011a), and the first methionine is numbered as Met0.

Yeast clone selection was performed on YPD plates containing 2,000 µg/ml Zeocin (Öberg et al, 2011a), and small-scale protein production was tested for selected strains. Membrane fractions were analyzed by SDS–PAGE and immunoblot. Strains with the highest production of His-tagged proteins were picked for large-scale production in 3-liter fermentors. For untagged-cpAQP1aa-243,

the strain corresponding to the largest colony on the high-Zeocin plate was used for fermentation directly. Protein production was induced by methanol fed-batch, typically lasting between 24 and 48 h, resulting in a yield of more than 250 g wet cells per liter culture. Cells were harvested by centrifugation (6,000*g*, 45 min, 4°C) and stored at −20°C.

### Membrane preparation, solubilization, and protein purification

For cpAQP1aa-FL, 60 g of cells were thawed at room temperature and resuspended in 200 ml breaking buffer (50 mM $KH_2PO_4$ [pH 7.5] and 5% [wt/vol] glycerol), supplemented with Complete EDTA-free protease inhibitor (Roche). A Bead Beater (BioSpec) was used to break the cells with 0.5-mm glass beads (Scientific Industries) grinding for 12 × 30 s, with 30-s cooling between runs. Cell debris was separated by a first centrifugation (10,000*g*, 30 min, 4°C) followed by a second centrifugation (15,000*g*, 30 min, 4°C). The crude membrane was harvested from the supernatant by ultracentrifugation (100,000*g*, 90 min, 4°C). The resultant membrane was washed with urea buffer (4 M urea, 5 mM Tris–HCl [pH 9.5], 2 mM EDTA, and 2 mM EGTA) and ultracentrifuged again at 158,000*g* for 90 min at 4°C. A final wash was performed to remove traces of urea by homogenizing the membrane in membrane resuspension buffer (20 mM Tris–HCl [pH 8.3], 300 mM NaCl, 10% glycerol, and 2 mM β-mercaptoethanol [β-MeOH]) and centrifuging at 158,000*g* for 1 h at 4°C. Finally, the membrane was homogenized in the membrane resuspension buffer and stored at −80°C. For untagged-cpAQP1aa-243 and His-tagged-cpAQP1aa-243, the same procedure was used except that the pH of the membrane resuspension buffer was adjusted to eight.

Membrane solubilization of cpAQP1aa-FL was carried out by mixing the membrane with solubilization buffer (20 mM Tris–HCl [pH 8.3], 300 mM NaCl, 10% glycerol, 4% [wt/vol] n-nonyl-β-D-glucopyranoside [NG, Anatrace], and 2 mM β-MeOH) in a 1:1 volume ratio supplemented with Complete EDTA-free protease inhibitor (Roche). The mixture was gently rocked for 1 h in a cold room. Unsolubilized material was removed by ultracentrifugation (149,000*g*, 30 min, 4°C), and the supernatant was transferred to a falcon tube containing pre-equilibrated Ni-NTA agarose beads (QIAGEN) and rocked for 3 h or overnight. 5 mM imidazole was added 20 min before centrifugation to avoid nonspecific binding. The supernatant containing unbound proteins was discarded after centrifugation at 3,000*g* for 3 min at 4°C. Nonspecifically bound proteins were removed by washing the matrix in 15 ml wash buffer (20 mM Tris–HCl [pH 8.3], 300 mM NaCl, 10% glycerol, 0.4% NG, 50 mM imidazole, and 2 mM β-MeOH) for three times, with centrifugation between washes (3,000*g*, 3 min, 4°C). Thereafter, the resin with bound protein was transferred to a 6-ml gravity-flow column (QIAGEN). Protein was eluted with 10 ml elution buffer (20 mM Tris–HCl [pH 8.3], 300 mM NaCl, 10% glycerol, 0.4% NG, 350 mM imidazole, and 2 mM β-MeOH). Protein fractions were analyzed using SDS–PAGE. Protein-containing fractions were pooled together, and buffer was exchanged to gel filtration buffer (20 mM Tris–HCl [pH 8.3], 100 mM NaCl, 10% glycerol, 0.4% NG, and 2 mM β-MeOH) immediately using a PD-10 column (GE Healthcare). The protein eluted from the column was concentrated to 500 µl using a 50-kD cut-off concentrator (Merck Millipore), filtered through a

0.2-µm centrifugal filter (VWR), and injected to a Superdex 200 increase gel filtration column (GE Healthcare), pre-equilibrated with gel filtration buffer. Fractions were analyzed using SDS–PAGE, and the protein-containing fractions were pooled together. For crystallization, the protein was concentrated to ~18 mg/ml and stored at −80°C. For His-tagged-cpAQP1aa-243, the protocols were same except that the pH of all the buffers was eight.

For the untagged protein, the solubilization buffer contained 20 mM Tris–HCl (pH 8), 10% glycerol, 4% NG, and 2 mM $\beta$-MeOH. After solubilization and centrifugation, the solubilized material was diluted in a 1:3 (vol/vol) ratio with dropwise addition of pre-chilled dilution buffer (20 mM MES [pH 6], 10% glycerol, 0.4% NG, and 2 mM $\beta$-MeOH). The diluted sample was filtered using a 0.2-µm bottle top filter (Thermo Fisher Scientific) and applied to a Resource S cation exchange column (GE Healthcare) pre-equilibrated with buffer A (20 mM MES [pH 6], 37.5 mM NaCl, 10% glycerol, 0.4% NG, and 2 mM $\beta$-MeOH) at a flow rate of 0.5 ml/min. After washing the column with 10 column volume of buffer A, protein was eluted with the gradient 0–40% of buffer B (20 mM MES [pH 6], 1 M NaCl, 10% glycerol, 0.4% NG, and 2 mM $\beta$-MeOH). Fractions were analyzed using SDS–PAGE, and the protein-containing fractions were pooled together and concentrated to 500 µl and injected to a Superdex 200 increase column (GE Healthcare) pre-equilibrated with gel filtration buffer (20 mM Tris–HCl [pH 8], 100 mM NaCl, 10% glycerol, 0.4% NG, and 2 mM $\beta$-MeOH). The untagged protein was analyzed, concentrated, and saved as described above. The identity of all proteins was verified by mass spectrometry.

### Optimization of the production of *cpAQP1* in *P. pastoris*

The full-length cpAQP1aa was successfully produced to high yields in the *P. pastoris* membranes, with an estimated yield of 63 mg highly pure protein per liter culture after a two-step purification procedure. This protein was used for functional characterization after reconstitution into liposome using stopped flow. Crystallization attempts were also pursued, but no crystals were achieved. In the optimization of the crystallization approach for cpAQP1aa, truncated versions of the protein were evaluated, where truncations were based on the alignment with vertebrate AQPs with available structures. The N-terminus of cpAQPaa is very short, and the focus for truncations leaned on the C-terminus of the protein which specifically aligned with the C-terminal domain of mammalian AQP0, 1, 2, and 5 (see Fig S1). Based on this alignment, six different C-terminal truncations were made from cpAQP1aa residue 228–243, where the shortest truncation showed the highest production level (38 mg/l), and it was therefore taken further to large-scale production and crystallization. Resolution down to 7 Å was achieved for this truncated version of cpAQP1aa, and based on inhomogeneity and instability during purification, three additional versions of the same construct were evaluated including an untagged variant because it was suspected that the His-tag had a negative influence on the crystallization event. This hypothesis turned out to be valid; a monodisperse peak appeared after size-exclusion chromatography, indicating a homogenous population of stable protein. Importantly, highly ordered crystals were achieved for this version, and a diffraction at 1.9 Å was achieved (Table S1).

### Phosphostain

The aquaporin from the climbing perch, cpAQP1aa-243, was analyzed for phosphorylation together with a positive and negative control, lysozyme, and the periplasmic chaperone spy, respectively, by using a selective phosphate-binding tag molecule (BTL-111), with streptavidin-conjugated HRP bound to it (prepared according the manual from Fujifilm Wako Pure Chemical Corporation). An SDS–PAGE is performed by loading 7.5 µg of each protein species and operated at 300 V for 15 min. After transferring the proteins to a PVDF membrane, the instruction manual is followed, which briefly consists of a soaking step with 1 × TBS-T for at least 1 h, 30 min of incubation with the tag molecule and consecutive washing steps. The blot was developed by using 4 ml of Genescript LumiSensor chemiluminescent HRP substrate.

### X-ray crystallography

Crystallization trials were carried out for all the constructs, and the best crystals were obtained for untagged-cpAQP1aa-243 by the hanging-drop vapor diffusion method at 4°C. The reservoir solution contained 100 mM Tris–HCl (pH 7.8), 5% poly-γ-glutamic acid low–molecular-weight polymer (γ-PGA, Na+ form, LM) and 30% (vol/vol) PEG400. Before setting up the crystallization drops, 4 µl of the reservoir solution was mixed with 1 µl 30% (wt/vol) D-sorbitol. For the crystals at lower pH, the reservoir solution contained 300 mM lithium sulfate, 100 mM ADA (pH 6.5), and 30% (vol/vol) PEG400; and 4 µl of the reservoir solution was mixed with 1 µl 30% (wt/vol) dextran sulfate sodium salt (Mr 5,000). Crystallization drops were then set up by mixing the reservoir/additive mixture with protein at 1:1 or 2:1 ratio, and the drops were left to equilibrate against 0.5 ml reservoir at room temperature, and crystals grew in cold room (4°C). Crystals with different shapes appeared after a few days. Crystals were mounted in cryoloops, flash frozen in liquid nitrogen, and stored for data collection.

### Data collection and structural determination

Complete data sets were collected at beam line P13 (wavelength 0.8 Å) at the PETRA III storage ring (DESY) (Cianci et al, 2017) and at beam line ID30B (wavelength 0.911647 Å) at European Synchrotron Radiation Facility (Svensson et al, 2015), France, for the low pH crystals. Data sets were processed using MOSFLM (Leslie, 1992) and scaled using SCALA of the CCP4 program suite (Collaborative Computational Project Number 4, 1994). Molecular replacement was carried out using the program Phaser from the CCP4 program suite (Collaborative Computational Project Number 4, 1994). A homology model of cpAQP1aa based on the crystal structure of bovine AQP1 (PDB: 1J4N) (Sui et al, 2001) was generated using the SWISS-MODEL web server (Biasini et al, 2014). The model was subjected to iterative rounds of model building and refinement in COOT (Emsley & Cowtan, 2004) and Phenix (Afonine et al, 2012). The final models consists of residues 1–34, 38–108, 114–226, and 124 water molecules for the structure at pH 7.8 and residues A2-34, 38–110, and 112–225; residues B4-34, 38–110, and 112–225; residues C4-226; and residues D1-34, 37–226 for the structure at pH 6.5. All structure figures were made in PyMOL (The PyMOL Molecular Graphics System, Version 2.3.1, Schrödinger, LLC). All MD simulations were conducted

using the MD engine (Krieger & Vriend, 2015) in the YASARA program (Krieger & Vriend, 2014), with the AMBER14 force field (Maier et al, 2015).

### Water and glycerol permeability studies

Water permeability studies were carried out using similar protocols described in Öberg et al (2011b) and Isaksson et al (2017). Liposome stock (25 mg/ml) was prepared by homogenizing *Escherichia coli* polar lipid extract (Avanti Polar Lipids) in reconstitution buffer (50 mM NaCl and 50 mM Tris–HCl [pH 8.3] for cpAQP1aa-FL and pH 8.0 for His-tagged-cpAQP1aa-243). An n-octyl-$\beta$-D-glucoside ($\beta$-OG; Anatrace) stock solution (50 mM Tris–HCl [pH 8.0], 50 mM NaCl, and 10% $\beta$-OG [wt/vol]) was added to the liposome solution to a final concentration of 1%. Purified protein was added into the mixture with a theoretical lipid-to-protein ratio of 50. Samples were gently mixed and incubated for 10 min. Subsequently, Biobeads (Bio-Rad Laboratories) was added to the mixture to remove the detergents by incubation (and rocking) at room temperature in dark overnight. The resulting proteoliposome was pelleted (100,000g, 1 h, 10°C) and gently resuspended in reconstitution (same as described before) buffer to a final lipid concentration of 2 mg/ml. Samples were filtered using a 0.2-$\mu$m spin column filter (VWR) by centrifugation (10,000g, 3 min, 20°C) before measurements. Control liposomes were prepared using the same protocol but with addition of corresponding gel filtration buffer instead of purified protein.

Water flow experiments were conducted on an SFM 2000 (Bio-Logic Science Instruments). In brief, 88 $\mu$l of sample was rapidly mixed with the same amount of hyperosmolar solution (reconstitution buffer with 300 mM sucrose). Change in light-scattering intensity at the wavelength of 438 nm was monitored at the fixed angle of 90°. At least two independent proteoliposome preparations were made for each target protein, and three samples were measured for each. For each sample, three measurements were averaged for each curve, providing the raw data for the calculation of the initial rate. The light-scattering data were normalized and fitted to a two-exponential equation:

$$I = A_1 \, e^{-k_1(t-t_0)} + A_2 \, e^{-k_2(t-t_0)} + A_3 \qquad (1) \, ,$$

where $I$ is the intensity corresponding to the time point $t$; $A_1$, $A_2$, and $A_3$ are coefficients; $k_1$ and $k_2$ are the rate constants of water flow; and $t_0$ is the dead time of the measurements.

Averaged curves (n = 6–31 [2–11]), where the number of individual preparations are given in parenthesis, were used to calculate the initial rate. Average of the initial rate, derived from the two-exponential equations, plus SD was the calculated for each target. Figures were made using Graphpad Prism version 6.0, and the error bars represent SD. The amount of protein in the liposomes was estimated using immunoblots, and the intensity of the immunoblot signal was quantified for each protein target (n ≥ 2). From this, a normalization factor relative full-length cpAQP1aa was calculated and used to calculate normalized k-values for all protein targets. The relationship between protein amount in the protein liposome and the initial rate is not necessarily linear, but the correlation gives an indication of the real differences in water flow rate for each mutant. Statistical significance was evaluated using unpaired $t$ test, and results shown in the bar graphs as stars together with the two tailed $P$-value for each analysis.

Glycerol flow measurements were performed using the same setup with the difference that the liposomes were preloaded with glycerol buffer (300 mM glycerol, 50 mM Tris [pH 8.3], and 50 mM NaCl), and the flow of glycerol out of the proteoliposome was measured via light scattering at 90° and a wavelength of 438 nm. For each measurement, 88 $\mu$l glycerol buffer and 88 $\mu$l sucrose buffer (300 mM sucrose, 50 mM Tris [pH 8.3], and 50 mM NaCl) were introduced, assuring that the osmolarity of 528 mOsm was applied. For each experiment, a minimum of three stopped-flow measurement traces were averaged and fitted to a two-exponential function using $R^2$ values of at least 0.95.

### Molecular dynamics simulations

All MD simulations were conducted using the MD engine (Krieger & Vriend, 2015) in the YASARA program (Krieger & Vriend, 2014), with the AMBER14 force field (Maier et al, 2015). The crystal structure of the aquaporin was first protonated assuming a pH of 7 and energy-minimized. The different mutants or phosphorylated variants (pY107, pT38, Y107A, L117A) were generated by modifying the corresponding residue of the minimized structure in silico, followed by additional energy minimization. Each system was placed in a cubic water box extending 10 Å away from nearest protein atom in each direction, water molecules were randomly replaced by Na$^+$ or Cl$^-$ ions to ensure charge neutrality, and assuming a physiological salt concentration of 0.9% NaCl.

Following a stepwise sequence of equilibration, the 100-ns production runs were performed on wild-type crystal structure and the five modified proteins, assuming T = 298 K, as maintained through velocity rescaling, and simulation box rescaling to ensure water density of 0.997 for pressure control. Each simulation box contained in the order of 55,000 atoms. All simulations were performed using the default macro md.run in the YASARA program. Analyses were performed with the macro md_analyze in the YASARA program, to which analysis of the bond distances were added.

## Data Availability

The atomic coordinates and structure factors (PDB ID: 7W7S and 7W7R) have been deposited in the Protein Data Bank (http://wwpdb.org/). Structures and trajectories from MD simulations are available at https://zenodo.org/ with DOI: 10.5281/zenodo.7094143.

## Supplementary Information

## Acknowledgements

The work presented in this study was supported by research grants from the Swedish research Council (K2013-66X-20431-07-05) to K Hedfalk; from the Swedish Research Council (2009-00360 and 2013-05945) to S Törnroth-

Horsefield; Academic Research Grant, Ministry of Education, Singapore (R154-000-632-112) and from the Environment and Water Industry Development Council, Singapore (R706-000-041-279), to K Swaminathan; and from the Swedish Research Council FORMAS (2012-771) to M Andersson and S Isaksson. We thank Zhengjun Li and Siew Hong Lam of Department of Biological Sciences, National University of Singapore, for providing the cpAQP1 plasmid, which we later modified and recloned. Matthijs Panman has been of crucial help in improving the quality of the script for analysing the stopped-flow data and Darius Sulskis provided the SPY protein. We thank the staff of the beam line P13 at the PETRA III storage ring (DESY, Hamburg, Germany) and beam line ID30B at ESRF, France for their help in data collection.

## Author Contributions

J Zeng: data curation, formal analysis, supervision, validation, investigation, methodology, and writing—original draft, review, and editing.

F Schmitz: data curation, software, formal analysis, supervision, validation, investigation, visualization, methodology, and writing—original draft, review, and editing.

S Isaksson: data curation, software, formal analysis, supervision, methodology, and writing—review and editing.

J Glas: data curation, formal analysis, validation, investigation, methodology, and writing—review and editing.

O Arbab: data curation, formal analysis, and methodology.

M Andersson: resources, supervision, funding acquisition, investigation, and writing—review and editing.

K Sundell: conceptualization, supervision, funding acquisition, investigation, and writing—review and editing.

LA Eriksson: conceptualization, resources, data curation, software, formal analysis, funding acquisition, validation, investigation, visualization, methodology, and writing—original draft, review, and editing.

K Swaminathan: conceptualization, resources, supervision, funding acquisition, investigation, and writing—review and editing.

S Törnroth-Horsefield: conceptualization, resources, data curation, software, formal analysis, supervision, funding acquisition, validation, investigation, visualization, methodology, and writing—original draft, review, and editing.

K Hedfalk: conceptualization, resources, data curation, software, formal analysis, supervision, funding acquisition, validation, investigation, visualization, methodology, project administration, and writing—original draft, review, and editing.

## Conflict of Interest Statement

The authors declare that they have no conflict of interest.

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
