## [Reviewer comments · Life Science Alliance]

High resolution structure of a fish aquaporin reveals a novel extracellular fold

Kristina Hedfalk, Jiao Zeng, Florian Schmitz, Simon Isaksson, Jessica Glas, Olivia Arbab, Martin Andersson, Kristina Sundell, Leif Eriksson, Kunchithapadam Swaminathan, and Susanna Törnroth-Horsefield

DOI: <https://doi.org/10.26508/lsa.202201491>

Corresponding author(s): *Kristina Hedfalk, University of Gothenburg and Susanna Törnroth-Horsefield, Lund University*

Review Timeline:

Submission Date:	2022-04-20
Editorial Decision:	2022-05-30
Revision Received:	2022-08-24
Editorial Decision:	2022-09-16
Revision Received:	2022-09-21
Accepted:	2022-09-22

Scientific Editor: Novella Guidi

Transaction Report:

May 30, 2022

Re: Life Science Alliance manuscript #LSA-2022-01491-T

Dr. Kristina Hedfalk
University of Gothenburg
Chemistry and Molecular Biology
Box 462
Göteborg 40530
Sweden

Dear Dr. Hedfalk,

Thank you for submitting your manuscript entitled "High resolution structure of a fish aquaporin reveals a novel extracellular fold" to Life Science Alliance. The manuscript was assessed by expert reviewers, whose comments are appended to this letter. We invite you to submit a revised manuscript addressing the Reviewer comments.

Thank you for this interesting contribution to Life Science Alliance. We are looking forward to receiving your revised manuscript.

Sincerely,

B. MANUSCRIPT ORGANIZATION AND FORMATTING:

Reviewer #1 (Comments to the Authors (Required)):

The authors resolved the structures of an aquaporin protein, cpAQP1aa, from an interesting species of fish, which is believed important for its osmoregulation and thus ability to survive both in water and on land. The structures were determined at pH6.5 (3.5Å) and pH7.8 (1.9 Å), both of which contained a novel conformation in the extracellular domain that the authors propose may function as a newly identified pH-insensitive gate of water-specific permeation. This, if proven, is a very interesting and novel finding in the regulatory mechanism of AQPs activity, and directly relates to the physiological function of the cpAQP1aa. However, several major issues need to be addressed before this conclusion can be made unambiguously.

Major points:

1. The claim that unique conformation of loop C leading to a lower water permeation activity of cpAQP1aa is not very well supported in the current manuscript. It seems residues near the extracellular gate, Y107, L117, etc., are conserved across cpAQP1aa and other AQPs, what is special in cpAQP1aa that resulted in the unique loop C conformation? Is the additional extracellular constriction in cpAQP1aa permeation pore shown in Fig. 2c caused by the L117 sidechain? If so, that the L117A mutation does not affect cpAQP1aa activity (Fig. 4a) suggests that this constriction is not likely related to gating. In addition, the authors found that the sidechain of Y107 is required to maintain the unique conformation of loop C and thus the lower activity of cpAQP1aa. However, Y107 is conserved (Fig. S1) across multiple AQPs including hAQP4, but hAQP4 exhibits higher activity (Fig. 4a) representative of a fully open pore (line 321). (It would also be nice to show the density maps of loop C since the author mentioned loop A and C densities are complete only in one of the protomers in the lower resolution structure (pH6.5) and are absent/incomplete in the higher resolution structure (pH7.8).)
2. No evidence has been presented to support that Y107 and T38 are indeed phosphorylation sites. Phosphorylation was observed neither in the structures nor the mass-spec experiments in this study (page 20). Are these sites within consensus sequences of known kinases? Are there other publications showing that these (or homologues etc.) are phosphorylation sites?
3. The claim that phosphorylation of Y107 and T38 regulates cpAQP1aa activity is not well supported. For T38, it would be nice to do T-tests between T38A vs T38E, and T38A vs WT. To my eye (Fig. 4a), T38A and T38E activities are similar, and both clearly higher than WT (cpAQP1aa-FL), in contrast to the authors' claim that T38E has higher activity but T38A is similar to WT (lines 332-334). For Y107E, Y107A is likely not a good mimetic of non-phosphorylated state since Alanine has a much smaller sidechain compared with Y. It would be nice to determine the activity of Y107F. In addition, the phosphorylated state mimic, Y107E, has very similar activity as WT. This could be because it is a poor mimic as the authors mentioned, but could also mean phosphorylation at this site does not affect function.
4. The pH in-sensitivity is not well supported. At pH6.5, cpAQP1aa vesicles show lower activity than hAQP4 (Fig. S5), and seem to have even lower permeability than empty vesicles in Figure S5d, leaving the possibility that lower pH may decreased cpAQP1aa activity. To evaluate the effect of pH on cpAQP1aa activity, it is preferable to use more stable proteoliposomes or perform cell-based experiment to enable measurements at both pH6.5 and pH7.8 for direct comparison.

Minor points:

- Page 6 line 76: By "membrane-bound" did you mean membrane-integral?
- Page 6 line 81: By "complexity" did you mean appeared later in evolution?
- Page 9 line 122: NPA motif full name not defined.
- Figure 2: Labeling motifs/loop names would be helpful to orient reader.
- Figure 2d: hydrogen bonds and density maps were in very similar color and hard to see.
- Figure 2e and f: is the view in f rotated relative to e, and showing different 'sticks'?
- Figure 2c: including color legend in pore radii plot would help to read.
- Page 9 line 124: "direct...specificity" or "determine...specificity"?
- Page 11 line 155: "AQP1" should cite the PDB ID of MR model.
- Page 15 line 231: Are loops A and C in protomer C in the pH6.5 crystal involved in crystal packing, resulting them to be more ordered than loops in other protomers?
- Page 17 line 287-289: Na⁺/H⁺ exchanger's role in ammonia permeation is purely speculative and not within focus of this study.
- Fig. 3b: Are densities of Arg200/187 good enough for distinguishing side chain conformations?

Reviewer #2 (Comments to the Authors (Required)):

The paper describes the first structure of a water channel from Fish. The crystal structure reveals that the main difference between the fish Aquaporin and previously determined Aquaporins is that the extracellular funnel is narrower. The authors attribute this difference the conformation of loop C. By phosphorylation mimic mutants measured by liposome swelling experiments and MD simulations they propose that loop C has evolved to only be more open once it phosphorylated. Previous, regulatory sites by phosphorylation have only been on the intracellular side. The reason for the extra regulation in the fish Aquaporin might be related to its physiology as this fish is able to adapt from fresh water to salt water.

Overall, I find this to be a solid paper and the liposome swelling experiments are appreciated. The final mechanism would naturally be supported by a direct confirmation of a phosphorylated loop C and subsequent liposome assays, but overall I think that there is enough new insights offered by the paper anyhow.

I think the text could be shortened in the introduction and discussion regarding fish physiology as, although interesting, does detract from the work.

Reviewer #3 (Comments to the Authors (Required)):

1. Summary

In this manuscript, Zeng and Hedfalk et al. report the crystal structures of a fish aquaporin protein. They found that cpAQP1aa is constricted partially by a loop at the extracellular port of the tunnel in a semi-open conformation, which is different from those found in other aquaporin structures. They used mutagenesis and MD simulation to propose a phosphorylation-dependent gating mechanism. This is an interesting study, which may provide molecular insight into the osmoregulation of the fish. The major issue of this work is that the mechanism and studies are based on an assumption that this Aqp is regulated by protein phosphorylation, which has never been experimentally verified or demonstrated. Therefore, the mechanism they proposed is interesting but highly speculative.

2. Major questions:

- 1) Is there any experimental evidence for the phosphorylation at Y107 and T38? Y107 is conserved in Aquaporins and T38 is also presented in some aquaporins based on Fig. S1. Is there evidence of phosphorylation at these two positions in other well-studied aquaporins?
- 2) The phosphorylation of SoPIP2;1 is located on the cytoplasmic surface, however, both Y107 and T38 are located on the extracellular surface. Although authors hypothesize that extracellular phosphorylation may occur, most protein kinases function in the cytoplasm. Extracellular protein phosphorylation is rare unless authors show direct evidence on cpAQP1aa. Their Mass Spec result showed phosphorylation occurs in other sites of the protein, not the sites they proposed when expressed in yeast.
- 3) Is it possible that the restricted pore contributes to the substrate selectivity, not gating? Authors showed SoPIP2;1 maintains high selectivity of water vs. glycerol, maybe ammonia. Any changes in substrate selectivity for the mutants, such as L117 or Y107?

3. Additional issue:

The manuscript may need a major revision to focus on the structural finding, not speculate on any unverified hypothesis.

Dear Prof. Guidi,

First of all I would like to express our gratitude for your time and consideration of our manuscript 'High resolution structure of a fish aquaporin reveals a novel extracellular fold' (Ref: #LSA-2022-01491-T) which we submitted to LSA the 20th of April 2022. We are very pleased to hear the overall positive comments from all three reviewers on the first aquaporin structure from fish, presented together with solid functional and mutational evaluation. We appreciate the chance to improve the manuscript by reflecting on reviewers' constructive comments. Below we have answered these comments, point by point. Mainly, we have restructured the manuscript so that the Result section focuses on the structural, functional, and mutational evaluation, leaving the discussion of a possible regulatory mechanism to the Discussion section. This is in good agreement with overall comments from all three reviewers. In addition, we have shortened the introduction and improved figures in line with reviewer's comments. All changes are highlighted in the manuscript file in red text (marked version of the resubmitted manuscript). Finally, the manuscript is reformatted to follow the style of Life Science Alliance.

Reviewer #1:

General comments:

The authors resolved the structures of an aquaporin protein, cpAQP1aa, from an interesting species of fish, which is believed important for its osmoregulation and thus ability to survive both in water and on land. The structures were determined at pH6.5 (3.5Å) and pH7.8 (1.9 Å), both of which contained a novel conformation in the extracellular domain that the authors propose may function as a newly identified pH-insensitive gate of water-specific permeation. This, if proven, is a very interesting and novel finding in the regulatory mechanism of AQPs activity, and directly relates to the physiological function of the cpAQP1aa. However, several major issues need to be addressed before this conclusion can be made unambiguously.

Reply:

We thank Reviewer #1 for his positive comments and for highlighting the strengths of the manuscript: the high-resolution structure of an aquaporin from an interesting species of fish, showing a novel conformation of the extracellular domain. We agree that, although there are several aspects of our results that support regulation by gating, this is still speculative. To clarify this, we have revised the manuscript so that the focus of the Result section lies on the structural, functional and mutational findings and the entire discussion about a possible gating mechanism has been moved to the Discussion section. For example, the heading of the fifth section in the Result section is changed from 'Possible regulatory mechanism for the cpAQP1aa involves the conformation the conformation of loop C and phosphorylation' to 'Mutational analysis of the extracellular constriction region'.

Major point 1:

The claim that unique conformation of loop C leading to a lower water permeation activity of cpAQP1aa is not very well supported in the current manuscript. It seems residues near the

extracellular gate, Y107, L117, etc., are conserved across cpAQP1aa and other AQPs, what is special in cpAQP1aa that resulted in the unique loop C conformation? Is the additional extracellular constriction in cpAQP1aa permeation pore shown in Fig. 2C caused by the L117 sidechain? If so, that the L117A mutation does not affect cpAQP1aa activity (Fig. 4A) suggests that this constriction is not likely related to gating.

Reply:

The reviewer is correct in that several of the residues around the extracellular constriction region are also found in other AQPs, including Leu 117 and Tyr 107. Nevertheless, loop C adopts a unique conformation in cpAQP1aa that is also stable in the molecular dynamics simulations (Fig. 4). This aspect is brought up in the Discussion, page 18; “Interestingly, Tyr 107 is rather well-conserved among fish species, as well as present in several mammalian AQP homologues, including AQP0, AQP4 and AQP7 (Fig. S1), However, in mammalian AQPs, the tyrosine side-chain adopts a different position, pointing away from the pore rather than towards it (Fig. 2F)”.

Furthermore, while the functional analysis of the Y107A mutant supports the role of the tyrosine side chain in stabilizing this conformation, L117A, as pointed out by the reviewer, behaves as wild-type. This suggests that it is the overall fold of loop C rather than the Leu side chain that restricts the pore, which we also state in the manuscript on page 13. This is supported by the structural analysis which show that the narrowest part of the channel is defined by backbone carbonyls, rather than side chains. To clarify this, we have added the following sentence on page 13-14: “This is in agreement with the structural analysis, which showed that the narrowest point of the channel is defined by the backbone carbonyls of Leu 117 and Gly and not the Leu 117 side chain 180 (Fig 2C)”.

Major point 2:

In addition, the authors found that the sidechain of Y107 is required to maintain the unique conformation of loop C and thus the lower activity of cpAQP1aa. However, Y107 is conserved (Fig. S1) across multiple AQPs including hAQP4, but hAQP4 exhibits higher activity (Fig. 4A) representative of a fully open pore (line 321).

Reply:

Y107 is indeed found in several aquaporin isoforms and it is not completely clear why it is able to adopt a unique conformation in cpAQP1aa that has not been observed in other AQP structures. Nevertheless, our combined structural and functional analysis clearly point at this residue being central for the hydrogen bonding network that stabilizes loop C in its unique conformation. In cpAQP1aa, as well as other fish AQPs, Tyr 107 is followed by a glycine, a residue that varies in other eukaryotic AQPs. It may be that it is the combination of these two residues that allows loop C to adopt the conformation seen in cpAQP1aa. To clarify this reasoning, the following paragraph is added to the discussion at page 18-19; “Furthermore, in cpAQP1aa, Tyr 107 is followed by a glycine, a residue that is highly conserved in fish but varies in other eukaryotic AQPs (Fig. S1). It may be that the combination of tyrosine and glycine at these positions is necessary for the unique conformation of loop C seen in cpAQP1aa. Furthermore, the conservation of the Tyr-Gly motif in fish suggests that the extracellular constriction region observed in cpAQP1aa may be a general feature amongst fish AQPs”.

(It would also be nice to show the density maps of loop C since the author mentioned loop A and C densities are complete only in one of the protomers in the lower resolution structure (pH6.5) and are absent/incomplete in the higher resolution structure (pH7.8).)

We have added the electron density map (composite omit map) to **Figure S4D**.

Major point 3:

No evidence has been presented to support that Y107 and T38 are indeed phosphorylation sites. Phosphorylation was observed neither in the structures nor the mass-spec experiments in this study (page 20). Are these sites within consensus sequences of known kinases? Are there other publications showing that these (or homologues etc.) are phosphorylation sites?

Reply:

The putative Y107 and T38 phosphorylation sites were identified using the GPS5.0 and this is now clarified in the manuscript on page 14. The server predicted them as being part of tyrosine kinase and CAMK/CAMK1 sites, respectively. A possible candidate for phosphorylation of Y107 is the secreted tyrosine kinase VLK which has been shown to phosphorylate similar sequences, see **Fig. S11**. The fact that VLK does not exist in yeast would explain why cpAQP1aa is not phosphorylated in *Pichia* during protein expression. For phosphorylation of Thr 38, a likely kinase candidate is less clear, but a member of the recently identified secreted kinase family FAM69 (DIA1) (Dudkiewicz M, et al. PLOS ONE, 2013, 8(6):e66427) is predicted to phosphorylate similar sequences (Hareza A et al., PeerJ. 2018 Apr 9;6:e4599. Similar to VLK, this kinase does not exist in yeast. These points are brought up in the Discussion, page 21 and 22, of the revised manuscript.

It should also be pointed out that aquaporins from fish are not well studied, and there are no publications on phosphorylation of these, nor any other particular sites. Hence, we see our study as a very interesting starting point for further analysis aiming towards a more thorough understanding of aquaporin regulation in fish.

Major point 4:

The claim that phosphorylation of Y107 and T38 regulates cpAQP1aa activity is not well supported. For T38, it would be nice to do T-tests between T38A vs T38E, and T38A vs WT. To my eye (Fig. 4A), T38A and T38E activities are similar, and both clearly higher than WT (cpAQP1aa-FL), in contrast to the authors' claim that T38E has higher activity but T38A is similar to WT (lines 332-334).

Reply:

We thank Reviewer #1 for pointing this aspect out and a more thorough statistical analysis is performed in the revised version of the manuscript. Comparing T38A versus T38E gives that they are not significantly different; a two-tailed P value equals 0.3318. Comparing T38A versus WT, on the other hand, gives a statistical difference and a two-tailed P value of 0.0064. The latter is now added to **Fig. 4A**. Hence, we cannot state that there is a difference between the two T38 mutants, wherefore we conclude that mutations at this specific residue influence the transport through the channel. The manuscript is updated with this reasoning, please see page 14.

Major point 5:

For Y107E, Y107A is likely not a good mimetic of non-phosphorylated state since Alanine has a much smaller sidechain compared with Y. It would be nice to determine the activity of Y107F. In addition, the phosphorylated state mimic, Y107E, has very similar activity as WT. This could be because it is a poor mimic as the authors mentioned, but could also mean phosphorylation at this site does not affect function.

Reply:

We agree with the reviewer that Y107A is not a good mimetic of a non-phosphorylated tyrosine. The mutant was primarily constructed to evaluate the role of the tyrosine side chain for the hydrogen bonding network that stabilizes loop C in its unique conformation. While Y107F may indeed be more suitable than Y107A for evaluating the non-phosphorylated state, we are confident that the recombinantly produced wild-type cpAQP1aa is not phosphorylated on Tyr 107 or Thr38, as shown by our mass spectrometry analysis (page 15). As such, we strongly believe that wild-type cpAQP1aa is the best representation of non-phosphorylated cpAQP1aa in our functional studies. This is now mentioned on page 15 of the revised manuscript.

Major point 6:

The pH in-sensitivity is not well supported. At pH6.5, cpAQP1aa vesicles show lower activity than hAQP4 (Fig. S5), and seem to have even lower permeability than empty vesicles in Figure S5D, leaving the possibility that lower pH may decreased cpAQP1aa activity. To evaluate the effect of pH on cpAQP1aa activity, it is preferable to use more stable proteoliposomes or perform cell-based experiment to enable measurements at both pH6.5 and pH7.8 for direct comparison.

Reply:

We acknowledge Reviewer #1 for highlighting the difficulty to perform the functional assay at lower pH and agree the experiment shown in Fig. S5D is not very conclusive. Since our conclusion on the putative pH effect is solely based on the structural evaluation, we have decided to remove this data and delete Fig. S5D from the revised version of the manuscript, as well as updating the corresponding text on page 12.

Minor points:

Page 6 line 76: By "membrane-bound" did you mean membrane-integral?
Page 6 line 81: By "complexity" did you mean appeared later in evolution?
Page 9 line 122: NPA motif full name not defined.

Many thanks for pointing these minor mistakes out, they are now all corrected.

Figure 2: Labeling motifs/loop names would be helpful to orient reader.

We have labelled the NPA- and ar/R regions in panel C and added loop names in panel E. We have also added additional labels to **Fig. 4D**.

Figure 2D: hydrogen bonds and density maps were in very similar color and hard to see.

We have improved the visibility of the hydrogen bonds by making them thicker and changing the colour to blue.

Figure 2E and F: is the view in f rotated relative to e, and showing different 'sticks'?

F is indeed rotated 90 degrees and we now show this in the figure as well as mention it in the figure legend. We have updated panel **E** and **F** so that they show the same residues in stick representation.

Figure 2C: including color legend in pore radii plot would help to read.

Colour legends have been added to this figure as well as to **Fig 4E**.

Page 9 line 124: "direct...specificity" or "determine...specificity"?

Page 11 line 155: "AQP1" should cite the PDB ID of MR model.

Many thanks for pointing these minor mistakes out, they are now all corrected.

Page 15 line 231: Are loops A and C in protomer C in the pH6.5 crystal involved in crystal packing, resulting them to be more ordered than loops in other protomers?

In monomer C, residues 124-127 of loop C are involved in crystal packing. This region is quite far away from the disordered region of loop C and has the same stable structure in both crystal structures. Hence, we do not expect this to cause the increased order of the proximal part of loop C.

Page 17 line 287-289: Na⁺/H⁺ exchanger's role in ammonia permeation is purely speculative and not within focus of this study.

This has now been deleted.

Fig. 3B: Are densities of Arg200/187 good enough for distinguishing side chain conformations?

Both the structures of AtTIP2;1 and cpAQP1aa are at high resolution and the density for the arginine is clear (for cpAQP1aa, see **Fig. 2D**)

Reviewer #2:

General comments:

The paper describes the first structure of a water channel from Fish. The crystal structure reveals that the main difference between the fish Aquaporin and previously determined Aquaporins is that the extracellular funnel is narrower. The authors attribute this difference the conformation of loop C. By phosphorylation mimic mutants measured by liposome swelling experiments and MD simulations they propose that loop C has evolved to only be more open once it phosphorylated. Previous, regulatory sites by phosphorylation have only been on the intracellular side. The reason for the extra regulation in the fish Aquaporin might be related to its physiology as this fish is able to adapt from fresh water to salt water.

Overall, I find this to be a solid paper and the liposome swelling experiments are appreciated. The final mechanism would naturally be supported by a direct confirmation of a phosphorylated loop C and subsequent liposome assays, but overall I think that there is enough new insights offered by the paper anyhow.

Reply:

We thank Reviewer #2 for the acknowledgement of the first aquaporin structure from a fish species with a putative new regulatory mechanism on the extracellular side of the membrane, possibly of relevance for the physiology of fish in general. We are also grateful for the recognition of the robust functional analysis using liposomes. We agree that the direct confirmation of phosphorylation is needed to verify our hypothesis, which is why the focus of the revised paper has been shifted towards the structural analysis and the discussion of regulatory aspects are collected in the discussion section.

Major comment 1:

I think the text could be shortened in the introduction and discussion regarding fish physiology as, although interesting, does detract from the work.

Reply:

We agree with Reviewer #2 and have shortened the introduction as well as the discussion regarding these aspects.

Reviewer #3:

General comments:

In this manuscript, Zeng and Hedfalk et al. report the crystal structures of a fish aquaporin protein. They found that cpAQP1aa is constricted partially by a loop at the extracellular port of the tunnel in a semi-open conformation, which is different from those found in other aquaporin structures. They used mutagenesis and MD simulation to propose a phosphorylation-dependent gating mechanism. This is an interesting study, which may provide molecular insight into the osmoregulation of the fish. The major issue of this work is that the mechanism and studies are based on an assumption that this Aqp is regulated by protein phosphorylation, which has never been experimentally verified or demonstrated. Therefore, the mechanism they proposed is interesting but highly speculative.

Reply:

We thank Reviewer #3 for the acknowledgement of an interesting study, and we agree that phosphorylation has to be proven in order to shed light on the molecular insight of the regulation. Nevertheless, our functional evaluation combined with MD analysis provides robust data for the discussion, and the revised version is edited to more clearly highlight this, while the Result section focuses on the structural evaluation. Please also see related comments to Reviewer #1 and #2 above.

Major comment 1:

Is there any experimental evidence for the phosphorylation at Y107 and T38? Y107 is conserved in Aquaporins and T38 is also presented in some aquaporins based on Fig. S1. Is there evidence of phosphorylation at these two positions in other well-studied aquaporins?

Reply:

To our knowledge, this is the first possible example of extracellular phosphorylation of an aquaporin, and also of a membrane channel in general. This aspect is brought up in the Discussion, on page 21; “The possible involvement of extracellular phosphorylation sites in the proposed cpAQP1aa gating mechanism is intriguing. All other known phosphorylation-dependent AQP gating mechanisms involve cytoplasmic phosphorylation sites (Nesverova & Tornroth-Horsefield, 2019) and, to our knowledge, this is the first time extracellular phosphorylation has been proposed to control gating of any membrane channel.

Please also see Replies to Major points 2 and 3 from Reviewer #1 above.

Major comment 2:

The phosphorylation of SoPIP2;1 is located on the cytoplasmic surface, however, both Y107 and T38 are located on the extracellular surface. Although authors hypothesize that extracellular phosphorylation may occur, most protein kinases function in the cytoplasm. Extracellular protein phosphorylation is rare unless authors show direct evidence on cpAQP1aa. Their Mass Spec result showed phosphorylation occurs in other sites of the protein, not the sites they proposed when expressed in yeast.

Reply:

While phosphorylation at cytoplasmic sites are indeed more common, extracellular phosphorylation is well established and has been shown to control membrane protein function (see Discussion, page 21). While direct evidence of phosphorylation *in vivo* would of course be ideal, we believe that our bioinformatic analysis nevertheless gives support for the possibility of these sites being phosphorylated. Specifically, we have identified two secreted kinase candidates, VLK and DIA1 that (1) is able to phosphorylate highly similar sequences, (2) is found in *A. testudineus*, and (3) is not found in yeast, explaining why these sites are not phosphorylated by the expression host. These points are explained in more detail in the manuscript on page 21 and 22 and also in our reply to Reviewer #1, Major comment 3.

Major comment 3:

Is it possible that the restricted pore contributes to the substrate selectivity, not gating? Authors showed SoPIP2;1 maintains high selectivity of water vs. glycerol, maybe ammonia. Any changes in substrate selectivity for the mutants, such as L117 or Y107?

Reply:

Thanks for pointing this interesting aspect out. We bring up the possibility of altered substrate selectivity on page 17 in the manuscript with regard to the MD simulation of pY107: "In addition, the pore widens at the ar/R-region due to the arginine being pulled away from the pore centre by its interaction with the Tyr 107 phosphate group, adopting a pore diameter more similar to aquaglyceroporins. These structural changes around the constriction and ar/R-regions suggest that phosphorylation of Tyr 107 could lead to altered solute permeability." These structural changes are not observed in the MD simulations of any of the mutants (L117A, Y107A, Y107E), suggesting that these do not have altered substrate selectivity. Additional liposome assays are planned in a follow up-study, where additional aspects will be investigated based on this finding.

Major comment 4:

The manuscript may need a major revision to focus on the structural finding, not speculate on any unverified hypothesis.

Reply:

We have carefully revised the manuscript to focus on the structural and functional analysis, leaving the possibility of a regulatory mechanism to the Discussion section. Although we agree that the mechanism is speculative, we think there is enough indirect evidence to present this as a hypothesis and hope that it will spur future work in the field of fish AQPs and osmoregulation.

We trust our revised manuscript meets with your approval and look forward to your response.

On behalf of all authors,

Yours sincerely,

Kristina Hedfalk, PhD

September 16, 2022

RE: Life Science Alliance Manuscript #LSA-2022-01491-TR

Dr. Kristina Hedfalk
University of Gothenburg
Chemistry and Molecular Biology
Box 462
Göteborg, Göteborg 40530
Sweden

Dear Dr. Hedfalk,

Thank you for submitting your revised manuscript entitled "High resolution structure of a fish aquaporin reveals a novel extracellular fold". We would be happy to publish your paper in Life Science Alliance pending final revisions necessary to meet our formatting guidelines.

- please note that LSA allows for only 2 corresponding authors total - 1 corresponding author and 1 secondary corresponding author
- please make sure to add the ORCID ID for both corresponding authors-you should have received instructions on how to do so
- please add the Twitter handle of your host institute/organization as well as your own or/and one of the authors in our system
- please add your supplementary figure legends and your supplementary materials and methods section to the main manuscript text; there is no word limit for this section
- please add a callout for Figure S10 E to your main manuscript text

Figure Check:

- should Supplementary Figure 11 really be a figure? if it possible please transform it in a table

A. FINAL FILES:

B. MANUSCRIPT ORGANIZATION AND FORMATTING:

Sincerely,

Reviewer #1 (Comments to the Authors (Required)):

The authors have restructured the manuscript to focus on their main findings and made improvements in the figures to make it easier to understand. Overall, my comments have been satisfactorily addressed. This work presents an interesting finding in how cpAQP1aa works. I have no further comments and think that this paper should be published.

Reviewer #3 (Comments to the Authors (Required)):

The issues have been addressed thoroughly in the revision.

September 22, 2022

RE: Life Science Alliance Manuscript #LSA-2022-01491-TRR

Dr. Kristina Hedfalk
University of Gothenburg
Chemistry and Molecular Biology
Box 462
Göteborg, Göteborg 40530
Sweden

Dear Dr. Hedfalk,

Thank you for submitting your Research Article entitled "High resolution structure of a fish aquaporin reveals a novel extracellular fold". It is a pleasure to let you know that your manuscript is now accepted for publication in Life Science Alliance. Congratulations on this interesting work.

DISTRIBUTION OF MATERIALS:

Again, congratulations on a very nice paper. I hope you found the review process to be constructive and are pleased with how the manuscript was handled editorially. We look forward to future exciting submissions from your lab.

Sincerely,
